# Histone Deacetylases: Molecular Mechanisms and Therapeutic Implications for Muscular Dystrophies

**DOI:** 10.3390/ijms24054306

**Published:** 2023-02-21

**Authors:** Martina Sandonà, Giorgia Cavioli, Alessandra Renzini, Alessia Cedola, Giuseppe Gigli, Dario Coletti, Timothy A. McKinsey, Viviana Moresi, Valentina Saccone

**Affiliations:** 1IRCCS Fondazione Santa Lucia, 00143 Rome, Italy; 2Unit of Histology and Medical Embryology, Department of Human Anatomy, Histology, Forensic Medicine and Orthopedics, University of Rome “La Sapienza”, 00161 Rome, Italy; 3Institute of Nanotechnology, National Research Council (CNR-NANOTEC), University of Rome “La Sapienza”, 00181 Rome, Italy; 4Institute of Nanotechnology, National Research Council (CNR-NANOTEC), 73100 Lecce, Italy; 5CNRS UMR 8256, INSERM ERL U1164, Biological Adaptation and Aging B2A, Sorbonne Université, 75005 Paris, France; 6Department of Medicine, Division of Cardiology and Consortium for Fibrosis Research & Translation, University of Colorado Anschutz Medical Campus, Aurora, CO 80045, USA; 7Department of Life Science and Public Health, Università Cattolica del Sacro Cuore, 00168 Rome, Italy

**Keywords:** histone deacetylase, muscular dystrophies, Duchenne Muscular Dystrophy, clinical trials

## Abstract

Histone deacetylases (HDACs) are enzymes that regulate the deacetylation of numerous histone and non-histone proteins, thereby affecting a wide range of cellular processes. Deregulation of HDAC expression or activity is often associated with several pathologies, suggesting potential for targeting these enzymes for therapeutic purposes. For example, HDAC expression and activity are higher in dystrophic skeletal muscles. General pharmacological blockade of HDACs, by means of pan-HDAC inhibitors (HDACi), ameliorates both muscle histological abnormalities and function in preclinical studies. A phase II clinical trial of the pan-HDACi givinostat revealed partial histological improvement and functional recovery of Duchenne Muscular Dystrophy (DMD) muscles; results of an ongoing phase III clinical trial that is assessing the long-term safety and efficacy of givinostat in DMD patients are pending. Here we review the current knowledge about the HDAC functions in distinct cell types in skeletal muscle, identified by genetic and -omic approaches. We describe the signaling events that are affected by HDACs and contribute to muscular dystrophy pathogenesis by altering muscle regeneration and/or repair processes. Reviewing recent insights into HDAC cellular functions in dystrophic muscles provides new perspectives for the development of more effective therapeutic approaches based on drugs that target these critical enzymes.

## 1. Overview of the Histone Deacetylases and Their Role in Striated Muscles

Transcriptional regulation in eukaryotes is strongly influenced by post-translational modifications (PTMs) of histones, the core proteins of chromatin, such as phosphorylation, methylation, and acetylation. Histone acetylation is probably the most well-characterized of these modifications, with hyperacetylation leading to an increase in gene expression, due to the relaxation of chromatin structure, while hypoacetylation has the opposite effect. The latter is mediated by histone deacetylases (HDACs) [1]. By doing this, HDACs influence the delicate balance between euchromatin and heterochromatin, thereby widely affecting gene expression in a prolonged fashion [2]. Therefore, the balance between the levels of histone deacetylation and acetylation plays a key role in the modulation of gene transcription and governs numerous developmental processes, being involved in the regulation of various genes associated with signal transduction, cell growth, and cell death, as well as disease states, including fluid and electrolyte disorders or cancers [3,4]. In addition, HDACs deacetylate non-histone proteins, such as p53 [5,6] as one of the first identified HDAC targets, thus regulating their activity.

The numerous HDACs have a wide range of expression and function in multiple cell types and tissues. In spite of the lack of complete knowledge of their roles, a global inhibition of deacetylase activity in the human body has been proposed as a therapeutical approach for various disease states, including muscle dystrophy. Before approaching this issue, it is therefore important to provide an overview of the HDAC family, focusing on the different roles HDACs play in striated muscle (Table 1). According to their sequence similarities with yeast orthologs and the use of either Zn^2+^ or NAD^+^ as cofactors [3,7,8], 18 human HDACs have been identified and grouped into four classes.

Class I HDACs shows similarity to the yeast deacetylase Rpd3p enzyme and include HDAC1, 2, 3 and 8. They are Zn^2+^-dependent, ubiquitously expressed enzymes, which are localized prevalently in the nucleus, playing a key role in the lysine deacetylation of N-terminal histone tails [9]. They are essential regulators of gene expression, being recruited to specific chromatin loci as a part of multi-protein complexes that control the acetylation state of histones and other chromatin-associated factors [10], resulting in chromatin condensation and transcriptional silencing [11]. The best-studied complexes include the NuRD, Sin3 and CoREST complexes, which contain HDAC1/2, and the SMRT/NCoR complex, which contains HDAC3 [12,13,14].

Thanks to tissue-specific knock-out (KO) mouse models, it has been established that HDAC1 and 2 often play redundant functions in the development or homeostasis maintenance of numerous tissues and cell types. In the heart, HDAC1 and 2 repress genes encoding contractile proteins and calcium channels [15], while, in skeletal muscle, they control autophagic flux and muscle metabolism [16].

HDAC3 is required for normal mouse development and tissue-specific functions by epigenetically controlling metabolism and circadian rhythms [17]. Cell type- or tissue-specific deletion reveal a role of HDAC3 in cardiac development and cardiomyocyte metabolism, since its absence leads to severe underdevelopment of the ventricular walls and to ventricular septal defects [18,19,20]. Of note, some of these functions in cardiac development are independent of its deacetylase activity; rather HDAC3 regulates gene transcription by recruiting other epigenetic factors to the NCOR complex [19] or by tethering peripheral heterochromatin to the nuclear lamina [18]. As a demonstration of the importance of HDAC3 in the whole-body metabolism, the deletion of *Hdac3* in skeletal muscle causes severe systemic and skeletal muscle-specific insulin resistance, impaired insulin and glucose tolerance, and diminished glucose uptake into skeletal muscle, overall impacting on muscle performance [21].

HDAC8 controls processes different from the other class I HDAC members and has a multifaceted role in human pathophysiology [22]. However, to date, no specific role in striated muscle has been reported.

Class II HDACs are similar to the yeast Hda1 deacetylase enzyme. This class is further subdivided into class IIa (HDACs 4, 5, 7 and 9) and class IIb (HDACs 6 and 10). While class IIa HDACs localize both in the nucleus and in the cytoplasm, class IIb HDACs are primarily in the cytoplasm and contain two catalytic sites. Class IIa HDACs are characterized by an extended N-terminal domain, containing conserved serine (Ser) residues, which are subjected to phosphorylation by several kinases, such as CaMK or SIK [23,24,25], facilitating HDAC nuclear export. Moreover, because of a Tyr-to-His mutation in their catalytic pocket, class IIa HDACs possess very low deacetylase activity compared to class I and class IIb HDACs [26]. This finding implies that class IIa HDACs regulate gene transcription via acting as scaffold proteins to recruit class I HDACs to specific genes, or as tethering proteins to anchor chromatin regions, or via sterically block transcription factor activity.

Among the members of class IIa, HDAC4 plays crucial functions in striated muscles. Increased expression of HDAC4 has been detected in skeletal muscle in different diseases, such as Duchenne Muscular Dystrophy (DMD) [27] and Amyotrophic Lateral Sclerosis (ALS) [28,29]: importantly, the observations in pre-clinical models have been validated in patients. Despite binding and repressing the activity of two major myogenic factors, i.e., MEF2 [30] and SRF [31], mice harboring a skeletal-muscle specific deletion of *Hdac4* are viable and do not display obvious defects in skeletal muscle [32]. While class IIa HDACs play redundant roles in the establishment of the metabolic pattern of skeletal muscle fibers, by repressing MEF2 [33], HDAC4 per se is necessary and sufficient to mediate responses upon different stimuli in skeletal muscle. For instance, deletion of *Hdac4* in differentiating skeletal muscle cells via myogenin:Cre recombinase, hampers muscle regeneration, due to the release of soluble factors that inhibit muscle precursor cell differentiation [34]; if the deletion of *Hdac4* occurs earlier in the myogenic cells, such as in Pax7^+^ cells, it compromises muscle stem cell (MuSC) proliferation and differentiation [35]. Together with HDAC5, HDAC4 connects neural activity to skeletal muscle transcription upon denervation, via both epigenetic regulation of gene expression [36,37,38] and by modulating nuclear and cytoplasmic non-histone protein acetylation [39,40], thereby mediating neurogenic muscle atrophy. Interestingly, deletion of *Hdac4* in skeletal muscle is protective in experimental models of neurogenic muscle atrophy in the early phases after the surgical procedure [37]; however, it resulted detrimental effects following long-term denervation, causing muscle degeneration due to the impairment in the activation of multiple signaling, including the oxidative stress response, the ubiquitin-proteasome system and the autophagic pathway [32]. Consistently, deletion of *Hdac4* in skeletal muscle in a mouse model of ALS worsened pathological features, advancing and exacerbating skeletal muscle atrophy and denervation by modulating several biological processes and gene networks [29]. Similar to ALS, HDAC4 expression is upregulated in DMD skeletal muscles [27], and, consistently, deletion of *Hdac4* results detrimental in both disease states. Indeed, *mdx* mice with a skeletal muscle-specific deletion of HDAC4 show increased muscle damage and hampered muscle regeneration, overall leading to decreased muscle function. HDAC4 prevalently localizes in the cytoplasm of dystrophic muscles, where it mediates activation of the membrane repair mechanism, likely through a deacetylase-independent activity, thereby affecting muscle necrosis, satellite cell survival and myogenic capacity [27]. Overall, these studies suggest that skeletal muscle up-regulates HDAC4 expression upon stress as a response to a disease state. In the heart, the N-terminal proteolytically derived fragment of HDAC4 finely regulates lipid metabolism and glucose handling through MEF2-dependent gene expression, ultimately protecting from heart failure [41,42].

HDAC5 acts as a negative epigenetic regulator of IL-6 synthesis and release in skeletal muscle, and *Hdac5* global KO mice show improved systemic glucose tolerance in response to exercise [43]. Of note, a non-deacetylase-dependent regulatory role of HDAC5 has been reported in cardiac cells. By using *Hdac5* global KO mice, it has been illustrated that HDAC5 is required for the interaction of the class I HDAC/Sin3 co-repressor complex with the Nkx2.5 and YY1 transcription factors and the consequent recruitment of the complex to promoter regions of either the Ncx1 or Bnp gene, which are important for cardiac hypertrophy [44].

HDAC9 is highly expressed in cardiac muscle even though it does not affect heart development. Nonetheless, mutant mice lacking *Hdac9* are sensitized to hypertrophic signals and exhibit stress-dependent cardiomegaly, suggesting that HDAC9 is a negative regulator of cardiomyocyte hypertrophy [45,46]. HDAC9 mutant mice showed an increase in slow fibers suggesting that its deletion results in enhanced slow-fiber gene expression [33].

Among class IIb HDACs, HDAC6 has been found associated with the class III deacetylase SIRT2 [47]. This complex interacts with poly-ubiquitin and poly-ubiquitinated proteins [48], and with tubulin and microtubules in the cytoplasm [49]. In particular, it has been observed that HDAC6 localizes at NMJs and its deletion protects against microtubule disorganization, markedly influencing NMJ structure [50]. Moreover, HDAC6 expression is upregulated during muscle atrophy, where it interacts with the E3-ubiquitin ligase MAFbx, participating to its activation; consistently, HDAC6 inactivation protects against muscle wasting in mice [51]. In the heart, HDAC6 was recently found to regulate myofibril stiffness and diastolic function [52].

HDAC10 mainly acts as polyamine deacetylase instead of lysine deacetylase [53], but its targets and functions in striated muscle are poorly characterized.

Class III HDACs show similarity to the yeast Sir2. In humans, the family consists of seven members, named sirtuins (SIRT1-7), whose activity depend on NAD^+^ [54]. *Sirt1* KO mice are sterile, smaller and present abnormalities in heart morphogenesis, due to p53 hyperacetylation and p53-dependent apoptosis [55]. Moreover, SIRT1 plays an essential role in the maintenance of mitochondrial integrity by modulating the MEF2 transcription factors in the heart [56]. Similarly to the yeast Sir2, SIRT1 exerts longevity effects against aging-associated pathologies, such as neurodegeneration, metabolic dysfunction [57], and cardiovascular diseases [58]. Consistently, SIRT1 levels decrease with age [59], promoting senescence. In skeletal muscle, SIRT1 inhibits FoxO1 and FoxO3 activity upon fasting, thereby protecting muscle from atrophy while promoting growth [60]. Importantly, SIRT1 is a sensor of energy metabolism, being triggered by AMPK, and deacetylates, thus activating, peroxisome proliferator activated receptor gamma coactivator 1α (PGC-1α) [61].

SIRT2 plays a pivotal role in regulating the whole-body metabolism. Upon high-fat condition, deletion of *Sirt2* reduces muscle insulin sensitivity and contributes to liver insulin resistance in mice, potentially by affecting mitochondrial acetylation state [62], although opposite data were previously reported in vitro in muscle cells [63]. Moreover, SIRT2 inhibition impairs myoblast fusion [64] and promote autophagic flux [65]. Consistently, SIRT2 activation protects myotubes against dexamethasone-induced atrophy through inhibition of the autophagy system [65]. Similarly, in the heart, SIRT2 overexpression protects from Ang II-induced cardiac hypertrophy and fibrosis, promoting AMPK activation by deacetylating the kinase LKB1 [66,67].

SIRT3 localizes in the mitochondria and it functions to maintain mitochondrial homeostasis under stress, having as a target at least one fifth of all mitochondrial proteins and regulating their activity [68]. In skeletal muscle, SIRT3 expression is downregulated in diabetic or high-fat diet fed mice and, conversely, it is upregulated upon fasting [69]. Mice lacking SIRT3 show decreased oxygen consumption and simultaneous increase in reactive oxygen species production, as well as higher oxidative stress in muscle, overall impacting the insulin signaling [69]. Thus, SIRT3 is an additional, potential therapeutic target for regulating skeletal muscle insulin sensitivity.

SIRT4 shows a mitochondrial localization and is a lysine deacylase that controls insulin secretion: *Sirt4* KO mice progressively develop glucose intolerance and insulin resistance, highlighting the importance of this mitochondrial enzyme in regulating leucine metabolism [70]. Indeed, *Sirt4* KO mice display deregulated lipid metabolism, leading to increased exercise tolerance and protection against diet-induced obesity, with elevated levels of malonyl CoA decarboxylase and decreased malonyl CoA in skeletal muscle and adipose tissues [71,72].

SIRT5 localizes in mitochondria and catalyzes the removal of PTMs on lysine residues, such as succinylation, malonylation, and glutarylation, thus regulating the activity of numerous enzymes involved in cellular metabolism, including fatty acid oxidation [73,74], ammonia cycle [75,76], ketogenesis [76] or respiratory chain [77] and redox metabolism [78]. SIRT5 deficiency suppresses mitochondrial ATP production and promotes AMPK activation in response to energy stress, which is sufficient to prevent left ventricular dilation and cardiac dysfunction in mice subjected to transverse aortic constriction [79].

SIRT6 localizes in the nucleus and has deacetylase [8], defatty-acylase [80], and mono-ADP-ribosylation [81] activities, playing important regulatory roles during physiological and pathological processes. In skeletal muscle, SIRT6 deficiency induces a reduction of AMPK activity and impaired glucose homeostasis and insulin sensitivity, leading to attenuated whole body energy expenditure, and weakened exercise performance [82]. In the heart, SIRT6 binds to and represses the promoter of IGF signaling–related genes, thereby acting as a negative regulator of cardiac hypertrophy. Consistently, SIRT6 KO mice develop cardiac hypertrophy and heart failure at around 8–12 weeks of age [83].

Recent efforts have identified SIRT7 involvement in various cellular processes such as ribosome biogenesis [84], gene expression and cellular metabolism [85,86], by promoting glucose production, and mitochondrial homeostasis [87,88]. Deletion of *Sirt7* in mice leads to premature aging, progeroid phenotype, and lethal heart hypertrophy due to enhanced activation of p53 [89].

Class IV HDACs include only HDAC11, which presents similarities to both Class I and Class II. It has been found that HDAC11 promotes MuSC proliferation, by activating Notch signaling, and negatively affects skeletal muscle regeneration by reducing MyoD1 transcription [90,91,92]. HDAC11 deletion also increases the number of oxidative myofibers in skeletal muscle by promoting a glycolytic-to-oxidative muscle fibers switch [93]. In the heart, HDAC11 deletion improved several parameters of diabetes mellitus-associated cardiac injury, including oxidative stress, apoptosis, inflammation and cardiac function [94], pointing to HDAC11 suppression as a potential therapeutic target for treating such a cardiac condition.

**Table 1 ijms-24-04306-t001:** Histone deacetylase classification, localization, and functions in striated muscles based on KO mice.

HDAC Member	Localization	Experimental Model	Outcomes
HDAC1 and HDAC2 *	Nucleus	Cardiac-specific HDAC1&2 double KO Skeletal muscle-specific HDAC1&2 double KO	Neonatal lethality with cardiac arrhythmias [15] Perinatal lethality and progressive muscle degeneration [16]
HDAC3 *	Nucleus	Cardiac-specific KO Skeletal muscle-specific KO	Underdevelopment of ventricular walls and ventricular septal defects [18,19,20] Insulin resistance, diminished glucose uptake into skeletal muscle, and reduced muscle performance [21]
HDAC4 **	Nucleus and cytoplasm	Cardiac-specific KO mice Skeletal muscle-specific KO	Reduced exercise capacity, characterized by cardiac fatigue [41] Hampered responses following denervation [32,37], in ALS [29], upon injury [34,35], or in DMD [27]
HDAC5 **	Nucleus and cytoplasm	Global KO mice	Improved glucose tolerance in response to exercise [43]
HDAC9 **	Nucleus and cytoplasm	Global KO mice	Stress-dependent cardiomegaly [46] Enhanced slow fibers gene expression [33]
HDAC6 **^$^	Cytoplasm	Global KO mice	Protection against muscle wasting and microtubule disorganization [50,51] Increased myofibril stiffness [52]
SIRT1 ***	Nucleus	Global KO mice	Sterile, smaller and with abnormalities in the heart [55] Dilated cardiomyopathy [56]
SIRT2 ***	Cytoplasm and mitochondria	Global KO mice	Reduced muscle insulin sensitivity and increased liver insulin resistance [62]
SIRT3 ***	Mitochondria	Global KO mice	Decreased oxygen consumption and increased oxidative stress in skeletal muscle [69]
SIRT4 ***	Mitochondria	Global KO mice	Glucose intolerance and insulin resistance [70] Increased exercise tolerance and protection against diet-induced obesity [71,72]
SIRT5 ***	Mitochondria	Global KO mice	Decreased mitochondrial ATP production and increased AMPK activation in response to energy stress [79]
SIRT6 ***	Nucleus	Global KO mice Skeletal muscle-specific KO	Cardiac hypertrophy and heart failure at around 8–12 weeks of age [83] Reduced AMPK activity, impaired glucose homeostasis and insulin sensitivity [82]
SIRT7 ***	Nucleolus	Global KO mice	Prematurely aging, progeroid phenotype and lethal heart hypertrophy [89]
HDAC11 ****	Nucleus and cytoplasm	Global KO mice	Accelerated muscle regeneration [90,91] Increased the number of oxidative myofibers [93] Improved numerous parameters of diabetes mellitus-associated cardiac injury [94]

* Class I HDACs; ** Class IIa HDACs; **^$^ Class IIb HDACs; *** Class III HDACs; **** Class IV HDACs.

## 2. Histone Deacetylases in Muscular Dystrophies

Several HDAC isoforms have been implicated in skeletal muscle remodeling, both in physiological and pathological conditions [95,96]. Ample work revealed that HDACs exert pivotal roles in regulating fiber type specification [96], muscle fiber size and innervation [29,37,97], metabolic fuel switching [16,98,99], muscle development [100], insulin sensitivity and exercise capacity [69,101,102], thus contributing to the maintenance of skeletal muscle homeostasis. The evidence of a wide variety of HDAC functions in skeletal muscle led to an increasing interest to clarify their roles in skeletal muscle disorders [29,96], including muscular dystrophies (MDs).

MDs consist of a heterogeneous group of genetic disorders characterized by progressive weakness and degeneration of skeletal muscles resulting in impaired muscle function [103]. Traditionally classified by a patient’s clinical presentation, muscle group involvement, mode of inheritance, age of onset and overall disease progression, MDs have been linked to a variety of distinct single-gene mutations [104]. So far, molecular genetic mapping techniques have shown that MDs are caused by numerous mutations in several genes encoding structural and functional muscle proteins, resulting in degeneration or dysfunction of skeletal muscle [104].

The most severe and the most common adult form of MD is Duchenne Muscular Dystrophy (DMD), which affects 1 in 3500–6000 live male births, and is caused by the lack of functional dystrophin protein due to mutations in the dystrophin gene (*DMD*) [105]. The structural role of dystrophin is closely related to its centrality in assembling the sarcolemmal Dystrophin-Associated Protein Complex (DAPC), which provides the molecular link between the cytoskeleton and the extracellular matrix of skeletal myofibers [106,107]. Lack of dystrophin results in mechanical instability causing myofibers rupture during contraction. Moreover, being connected with multiple proteins, dystrophin modulates several signal transduction pathways, including Ca^2+^ entry, nitric oxide (NO), and reactive oxygen species (ROS) production [108,109,110]. The *mdx* mouse, harboring a nonsense point mutation in the exon 23 that aborts the full-length dystrophin expression, is the most widely used animal model for DMD research [111]. Despite the loss of dystrophin, *mdx* mice show minimal clinical features of the disease, if compared with DMD patients, probably due to compensatory mechanisms. The latter include muscle regeneration, which is more efficient in *mdx* mice compared to DMD patients, in part due to differences in telomere shortening and muscle stem cell regenerative capacity [112]. Among compensatory mechanisms triggered by the absence of dystrophin, the upregulation of *utrophin* has been reported in both DMD and *mdx* myofibers [113]. Utrophin is a structural and functional autosomal paralogue of dystrophin, normally located at the neuromuscular and myotendinous junctions in adult skeletal muscle in physiological condition [114], but enriched at the sarcolemma in dystrophic myofibers where it acts to preserving muscle function and mitigating necrosis [113]. Importantly, while the exogenous expression of *utrophin* attenuated the *mdx* dystrophic phenotype, its deletion in *mdx* mice worsened the pathology, thus confirming that utrophin protective functions in DMD [115,116,117].

In addition to dystrophin, another important member of the DAPC is the sarcoglycan complex, which is composed of four sarcoglycan (SG) proteins, α−, β−, δ−, and γ-SG, playing a key role to protect striated muscle membranes against contraction-induced damage [118,119]. Mutations in one of the four sarcoglycan genes (*SGCA*) cause a different form of autosomal recessive sarcoglycanopathies [120,121], a subgroup of Limb Girdle Muscular Dystrophies (LGMDs). Sarcoglycanopathies are more frequently found among the most severe forms of MDs, and the clinical phenotype closely resembles that of DMD, with onset during childhood [122,123].

The role of HDACs in MDs is not yet fully identified; indeed, most of our knowledge derives from studies with HDAC inhibitors in dystrophic contexts (discussed below). However, several studies revealed the deregulation of HDAC expression or activity in dystrophic muscles (Figure 1).

Higher global deacetylase activity was first detected in muscles of *mdx* mice and in DMD patients [124,125], accompanied by selectively elevated levels of HDAC2 in MuSCs [124]. Further investigations revealed a molecular link among the DAPC, NO signaling and HDAC2 [124,125]. Indeed, in *mdx* mice, the loss of an essential component of the dystrophin–glycoprotein complex leads to the displacement of the muscle-specific variant of the neuronal nitric oxide synthase (nNOSm) enzyme, which is normally located at the sarcolemma in close contact with the DAPC complex, resulting in reduced generation of NO. In addition to impairing many processes, including mitochondrial biogenesis and glucose metabolism, reduced NO bioavailability alters S-nitrosylation of HDAC2, resulting in increased activity and constitutive inhibition of HDAC2-target genes in dystrophic muscle [124,126,127]. HDAC2 directly inhibits *follistatin* gene transcription in *mdx* muscle cells, which in turn blocks a powerful inhibitor of muscle growth, i.e., *myostatin* [128]. Consistently, the follistatin-myostatin axis has been identified as a target to ameliorate MDs; indeed, myostatin blockade at early stages of the disease provides a beneficial effect in both *mdx* and α-SG–deficient mice [129,130]. Moreover, HDAC2 modulates a specific subset of miRNAs, including miR-1 and miR-29, while HDAC1 specifically inhibits miR-206 in dystrophic MuSCs, thereby correlating with several pathogenetic traits of DMD [127]. HDAC2 downregulation by siRNA or NO-donor led to improved myogenesis of *mdx* MuSCs in vitro, in addition to ameliorating functional and morphological parameters in vivo [124].

Among the members of class I HDACs, HDAC3 has been shown to be directly involved in the pathogenesis of the X-linked Emery–Dreifuss muscular dystrophy (EDMD1) [131,132]. This disease is caused by mutations in the *emerin* gene, which encodes for a nuclear membrane protein that binds to and recruits HDAC3 to the nuclear lamina. The loss of emerin in muscle cells leads to aberrant nuclear envelope architecture and heterochromatin organization, which results in a more open conformation because of the delocalization and loss of activity of HDAC3. As a result, skeletal MuSCs are unable to differentiate, resulting in progressive skeletal muscle wasting and impaired skeletal muscle regeneration [133]. Moreover, muscles from EDMD1 patients and *emerin*-null mice show an increased and improper temporal expression of marker genes involved in muscle regeneration, including Pax7, MyoD, and Myf5 [134]. Importantly, activation of HDAC3 catalytic activity by theophylline treatment rescues myogenic differentiation in *emerin*-null mice, confirming HDAC3 as a master regulator in coordinating the spatiotemporal localization of gene loci to the nuclear envelope required for proper differentiation and muscle regeneration [135].

In a recent study, HDAC8 was found to be overexpressed in DMD human primary myoblasts and myotubes, and in a zebrafish DMD model [136]. In the same study, the authors clarified the role of HDAC8 in modulating cytoskeletal architecture and stability through the deacetylation of α-tubulin. Moreover, selective inhibition of HDAC8, by PCI-34051 administration, rescues the DMD phenotype in terms of increased human myoblast differentiation and reduced lesion extent in zebrafish embryos, overall restoring skeletal muscle histomorphology and reducing inflammation [136].

Differently from class I HDACs, which predominantly localize to the nucleus, where they mostly act as epigenetic regulators, class IIa HDACs shuttle between the nucleus and the cytoplasm, regulating numerous stress responses. HDAC4 has been shown to be crucial for proper MuSCs proliferation and differentiation [35] and muscle regeneration [34] following acute muscle injury. A recent paper revealed enhanced expression of HDAC4 in *mdx* and DMD muscles, characterized by a higher cytoplasmic abundance of HDAC4 [27], thus suggesting a potential role for HDAC4 in this pathology. *Mdx* mice carrying a skeletal muscle-specific deletion of HDAC4 developed a more severe MD pathology, with increased muscle damage and reduced muscle regeneration, overall showing decreased muscle performance. The protective role of HDAC4 in the cytoplasm of dystrophic muscles is independent of its deacetylase activity and depends on its involvement in the membrane repair process. Indeed, cytosolic HDAC4 mediates the activation of a compensatory mechanism of membrane repair in *mdx* muscles, thus promoting MuSCs survival and differentiation, ultimately improving muscle regeneration and function [27].

HDAC5 is downregulated in the nucleus of *mdx* muscle and MuSCs, compared with normal controls, and has been implicated in the epigenetic control of chromatin landscape during *mdx* MuSCs differentiation. Impaired NO-dependent protein phosphatase 2A activity induces a hyperphosphorylation of HDAC5, thus reducing the amount of nuclear HDAC5 in complex with HDAC3, and affecting *mdx* MuSCs differentiation [125].

Regarding class IIb HDACs, two independent groups identified interesting functions for HDAC6 in DMD [137,138]. HDAC6 exclusively localized in the cytoplasm, where it removes acetyl groups from non-histone proteins such as α-tubulin, modulating microtubule network stability and organization [49]. HDAC6 also possesses a non-enzymatic zinc-finger ubiquitin-binding domain at its C-terminus, through which HDAC6 interacts with components of the ubiquitin proteasome pathway, thus playing a critical role in the cellular response to misfolded and aggregated proteins [139]. Moreover, HDAC6 and its endogenous inhibitor paxillin, regulate acetyl choline receptors (AChR) clustering at the neuromuscular junctions, by mediating a fine balance of nonacetylated and acetylated microtubule network [50]. Increased HDAC6 protein expression has been reported in *mdx* muscles, with a concomitant reduction of acetylated α-tubulin, which contributes to the disorganization of microtubule network and to the impairment of the autophagic flux in DMD. The pharmacological inhibition of HDAC6, by tubastatin A administration, restores the microtubule acetylation and rescues the autophagic flux enhancing autophagosome-lysosome fusion in *mdx* mice, in addition to improve AChR clustering and distribution [137,138]. Moreover, HDAC6 inhibition downregulates transforming growth factor beta (TGF-β) signaling, through an increase of SMAD2/3 acetylation, thereby reducing muscle atrophy and fibrosis and improving protein synthesis in *mdx* muscles [137].

Members of class III HDACs rely on NAD^+^ to deacetylate their targets, thereby mediating several important functions in skeletal muscle physiology and diseases [95,140]. Although SIRT1 expression does not change between *mdx* and control muscles, the lack of dystrophin abrogates proper diurnal oscillation of SIRT1 mRNA expression [141]. Moreover, an increased level of phosphorylated SIRT1 (p-SIRT1) was observed in *mdx* muscles, with a concomitant increase of histone H3 acetylation at Lys9/Lys14, thus suggesting attenuated SIRT1 activity [142]. In addition, NAD^+^ concentration was found to be reduced in dystrophic muscles, supporting a model in which SIRT1 activity is downregulated in *mdx* mice [143,144]. Functional proof that SIRT1 downregulation contributes to MD pathogenesis comes from gain-of-function studies. *Mdx* mice overexpressing SIRT1 in skeletal muscle developed a less severe DMD pathology, with decreased myofiber necrosis, oxidative stress and fibrosis, accompanied by a fast-to-slow myofiber shift, and overall improvement of muscle performance [143]. Most of the improvements reported in *mdx* SIRT1 overexpressing mice have been proven to be mediated by the activation of peroxisome proliferator-activated receptor, gamma, coactivator 1 alpha (PGC-1α), a SIRT1 target known to protect and ameliorate dystrophic muscles [145,146]. Indeed, increased expression of PGC-1α in dystrophic muscle mimics, in part, *mdx* SIRT1 transgenic mice, enhancing mitochondrial biogenesis, improving the oxidative metabolism and driving a fast-to-slow fiber switch, and preventing muscle degeneration [147]. Skeletal muscle-specific *Sirt1* knockout mice display a mild dystrophic phenotype, being more prone to suffer from exercise-induced muscle injury, probably due to defects in membrane resealing [148]. However, *Sirt1* loss in skeletal muscle of *mdx* mice does not exacerbate the dystrophic phenotype [148], suggesting redundant protective mechanisms in skeletal muscle under stress conditions.

SIRT2 modulates autophagy signaling, thereby affecting skeletal muscle atrophy and myoblast proliferation [65,149]. A recent role for SIRT2 in skeletal muscle following injury has been demonstrated. *Sirt2* KO mice showed a delay in muscle regeneration due to a decreased expression of anabolic and cell cycle regulators genes, with a concomitant increase in catabolic genes and muscle atrophy [150]. Interestingly, a significant upregulation of SIRT2 mRNA has been reported in MuSCs derived from DMD patients [151]. These recent results illustrate that further research is needed to better understand the role of SIRT2 in MDs, since SIRT2 could be a promising new therapeutic target in those muscular pathologies where regeneration is inefficient. Moreover, SIRT2 has been proposed as new serum dystrophic marker, since it is upregulated in *mdx* serum while it is reversed to control levels by overexpressing utrophin in *mdx* mice [152].

SIRT3, SIRT4, and SIRT5 are exclusively localized to mitochondria and regulate a wide range of metabolism-oriented enzymes in skeletal muscle, thereby modulating energy metabolism in response to mitochondrial stress. Mitochondrial dysfunction is a pathological feature of several MDs [153,154], suggesting a possible involvement of these sirtuins in such pathologies. SIRT3, SIRT4 and SIRT5 mRNA expression have been found to be upregulated in MuSCs derived from DMD patients and *mdx* mice [151], although no further investigations elucidating their potential role in MDs have been performed.

SIRT6 plays a pivotal role in heterochromatin stabilization through deacetylation of H3K9ac, H3K18ac and H3K56ac. In skeletal muscle, SIRT6 has been reported to negatively regulate *myostatin* expression via suppressing NF-*κ*B signaling, in addition to modulating glucose homeostasis and insulin sensitivity [82,155]. SIRT6 expression has been found to be upregulated in skeletal muscle and in MuSCs of *mdx* mice, where it mostly acts on H3K56ac, thereby repressing several genes, including *utrophin* and *myostatin* [151]. Lack of SIRT6 reduces muscle fragility and damaged myofibers, increasing the physical activity of *mdx* mice. Interestingly, *Sirt6*-depleted MuSCs showed attenuated activation, characterized by a strong reduction of Pax7/MyoD double-positive cells, reduced proliferation rate, and decreased expression of stress response-related genes [151]. Overall, these results indicate that reducing the persistent and chronic activation of MuSCs in *mdx* muscles is protective, and that inactivating SIRT6 in DMD ameliorates the dystrophic phenotype in mice.

The class IV HDAC11, which is a lysine de-fatty acylase [156,157,158], is highly expressed in skeletal muscle but it is dispensable for adult muscle growth. Interestingly, its genetic deletion accelerates regeneration in response to muscle injury [91,159]. The recent study on HDAC11-deficient mice show a more efficient muscle regeneration following acute injury [91] likely due in part to an increase in IL-10, which allows a faster transition from inflammatory to pro-regeneration environment. Since high levels of IL-10 have been demonstrated to ameliorate the pathology of *mdx* mice [160,161], these new results on HDAC11 functions are promising and open new avenues for the development of more specific HDAC inhibitors, such as specific HDAC11 inhibitors, as an effective approach to treat MDs. Further studies are needed to evaluate whether this HDAC is involved in the persistent and inefficient regeneration in MDs, and to verify whether HDAC11 is a candidate target to improve muscle repair in this pathological condition.

## 3. Targeting Histone Deacetylases in Muscular Dystrophies

Epigenetic mechanisms controlling transcriptional programs in tissue progenitors are becoming a critical area of interest in medicine. Indeed, current studies are focused on manipulating chromatin targets of individual signaling pathways to provide novel regenerative strategies based on epigenetic drug administration.

Numerous studies have highlighted the fundamental role of HATs and HDACs in regulating muscle gene transcription and therefore, muscle development and differentiation. Moreover, cumulative in vitro and in vivo evidence in the last years has underscored the link between HDAC deregulation and the pathogenesis of several MDs, in particular of the most severe one, the DMD [162,163,164]. In this context, HDACi have been shown to act in a selective way, potentiating myogenesis through the hyperacetylation of genes regulated during development and resolving their epigenetic bivalency, a characteristic signature that identifies genes poised for transcription that typically are enriched in embryonic stem cells or pluripotent cells [165]. Starting from this evidence, by inhibiting HDACs and reestablishing the epigenetic events necessary to activate adult stem cells, it represents one of the most powerful approaches to restoring the downstream networks of muscle regeneration and muscle homeostasis, leading to increased functional and morphological recovery of dystrophic muscles.

At first, focusing on the HDACi activity on skeletal muscle cells in vitro, it was observed that the pharmacological treatment targets myogenic differentiation [97,166]. Indeed, treatment of wild-type myoblasts with pan-HDACi, such as Trichostatin A (TSA), Valproic acid (VPA), or Sodium Butyrate (PhB), increases their differentiation potential and fusion capacity, due to different mechanisms: (i) the upregulation of MyoD acetylation; (ii) the modulation of histone acetylation at specific gene promoters and (iii) the increase of the expression of the pro-myogenic protein follistatin [97,166].

Several years ago, a link between dystrophin loss and HDAC activity was demonstrated [124,125]. In *mdx* whole muscles and primary myoblasts, an increase in global HDAC activity and HDAC2 expression was observed in association with a reduction in follistatin expression. Inhibition of HDAC2, by using the class I HDAC inhibitor MS-275 or siRNA, restores the level of global HDAC activity similar to healthy control muscles, leading to morphological and functional benefits in dystrophic muscles [124]. In more recent studies, increased activity of class I, class IIa and class I/IIb HDACs in muscles of 1.5-month-old *mdx* mice [27] and in Fibro-Adipogenic Progenitors (FAPs) isolated from 1.5 month- and 12 month-old *mdx* mice has been reported [167], further suggesting the involvement of HDACs in the pathogenesis of DMD.

Next-generation sequencing studies have focused on the fine regulation of myogenesis by HDACi, paying attention to the epigenetic players that create changes in the epigenome, opening new therapeutic options in muscle diseases. It emerged that most of the beneficial effects of the HDACi on dystrophic muscles arise from their ability to selectively activate a microRNA-SWI/SNF-based epigenetic network in FAPs, a specific population of mesenchymal cells resident in muscle interstitium [168,169]. FAPs are a muscle cell population that, while in regenerating conditions support MuSCs differentiation, in pathological conditions, such as DMD, contribute to the progression of the disease, affecting fibrotic and fat deposition, decreasing muscle contractility, and altering metabolism [170,171,172]. Intriguingly, pan-HDACi manipulate cell fate determination that redirects the lineage commitment of FAPs from a fibro-adipogenic toward a myogenic one [169].

In the context of MDs, it is worth mentioning the sirtuins, which are class III histone/protein deacetylases, are able to modulate several important physiological mechanisms such as inflammation, apoptosis, glucose homeostasis, life span, and neuroprotection. Acting pharmacologically on these enzymes permits modification of the acetylation state of several intracellular messengers, thereby regulating downstream mechanisms. This approach likely has strong therapeutic potential for many human diseases such as metabolic disorders, and degenerative diseases such as MDs. As described above, the most studied of the sirtuins is SIRT1, which is expressed in many tissues, including skeletal muscle and heart, where it deacetylates and activates PGC-1α, a key modulator of muscle metabolism. The activated form of PGC-1α controls mitochondrial biogenesis and homeostasis, and therefore SIRT1 modulation was seen to be associated with muscle pathologies. It is now well known that PGC-1α overexpression in dystrophic *mdx* mice leads to milder signs of pathology and an improved function both in normal condition and after intense physical exercise [61,145]. Other mechanistic roles are attributed to SIRT1 modulation, supporting the beneficial effects on muscle pathologies. It has been described for example that SIRT1 stimulates and restores autophagy in muscle tissue through the deacetylation of autophagy components, including Atg5, Atg7, and Atg8, and activating FoxO3a a transcription factor that regulates autophagy in skeletal muscle [173,174]. Moreover, SIRT1 may modulate the activity of SMAD transcription factors, key TGF-β signaling components that are involved in myofibroblast differentiation. The activity of SMAD is regulated by lysine acetylation/deacetylation, which plays a critical role in tissue fibrosis [175].

All these data generated in vitro on cells (Figure 2), together with the in vivo evidence of deregulated activity of HDACs in MDs, have provided the rationale for using pan-HDACi and modulators of sirtuins in preclinical studies, with the aim of assessing the ability of these classes of compounds to improve muscle regeneration and counteract muscle degeneration in models of MD.

### 3.1. Preclinical Studies

#### 3.1.1. HDACi in DMD

In DMD muscles, the lack of dystrophin, in addition to the events previously described, leads to a deregulation in the expression and in post-transcriptional modifications of all the DAPC components, causing a strong fragility of muscle fibers after contraction [176,177]. Muscle degeneration in turn activates compensatory regeneration to reduce the muscle damage. These processes characterizing DMD pathology, together with the increase of HDAC activity, led to the hypothesis that epigenetic treatments, based on pan-HDACi, could represent an encouraging approach to reduce the DMD progression by enhancing the formation of multinucleated myotubes and therefore of new muscle fibers [124,162,163].

The three pan-HDACi studied in vitro on wild-type myoblasts, TSA, VPA, and PhB, were also tested in vivo in *mdx* mice, by daily intraperitoneal injections in young *mdx* mice in which the compensatory regeneration phase is still active [178]. The results of this study established the ability of pan-HDACi to improve the differentiation potential of MuSCs after 10 days of treatment. However, long-term treatment of *mdx* mice for 3 months revealed that TSA represents the best epigenetic compound used to treat *mdx* pathology, among the three pan-HDACi used. These conclusions were drawn based on a decrease of DMD biomarkers and creatine kinase levels in blood, in addition to the absence of side effects. Moreover, histological and functional analyses revealed that TSA improves muscle force and muscle size, as well as reducing fibrotic scars and fat deposition, slowing progression of the disease. These effects were associated with upregulation of follistatin expression in *mdx* MuSCs [178]. Deepening the effect of pan-HDACi on *mdx* mice in vivo, further studies revealed a stage-specific effect of the epigenetic drugs. Indeed, it was demonstrated that the beneficial effects of TSA in *mdx* mice are restricted to the active regeneration window of time, while if the treatment starts at late stages of the disease, where the regeneration potential is exhausted, the compound loses its beneficial effects on muscle regeneration [111,168].

During the last decade, different pan-HDACi were tested on *mdx* mice (Figure 3 and Table 2). A dose-dependent study of suberoylanilide hydroxamic acid (SAHA) treatment demonstrated its effectiveness in ameliorating both dystrophic muscle function and morphology, reducing inflammation and fibrosis and also attenuating cardiac arrhythmias [179,180]; in addition, a preclinical dose-dependent study of ITF2357 (givinostat) highlighted its ability to recover muscle function and counteract muscle degeneration of *mdx* mice [163].

At the cellular level, the beneficial effects of pan-HDACi in *mdx* mice have been mainly associated with FAPs [168,169,181]. It was shown that HDACi treatment of young regenerating *mdx* mice exerts a double positive effect on *mdx* FAPs: first, it converts their lineage commitment toward a pro-myogenic one, and secondly, it stimulates their positive interaction with MuSCs, promoting muscle regeneration. In particular, pan-HDACi treatment controls dystrophic FAP lineage commitment, reducing their ability to contribute to fibrotic scar infiltration and adipocyte accumulation while inducing their latent myogenic phenotype, confirming that FAPs represent one of the cellular targets of pan-HDACi treatment [168,169]. Deciphering the molecular mechanisms behind this effect of pan-HDACi, it was demonstrated that the treatment changes chromatin structure at muscle loci of FAPs, inducing the expression of muscle genes, such as BAF60C and MyoD, and of muscle-specific miRNAs (myo-miRs), including miR-1.2, miR-133, and miR-206 [169,182]. These studies identified the HDAC–myo-miR–BAF60 network as up-regulated by pan-HDACi treatment in FAPs. Briefly, myo-miRs target and repress the expression of the alternative BAF60 variants of the SWI/SNF complex, BAF60A and BAF60B, responsible for the fibro-adipogenic phenotype, favoring the 172expression of the BAF60C variant, which in turn activates the transcription of muscle genes leading to a promyogenic commitment of FAPs. Such an effect, again, was not observed in FAPs from old *mdx* mice [169,182]. The inefficacy of pan-HDACi to ameliorate the dystrophic phenotype at late stages of the disease has been recently investigated by using genome-wide approaches, and it was found to be related to aberrant HDAC activity and to a senescent state of FAPs that is not reversed by the treatment [167]. Regarding the ability of FAPs to support MuSC differentiation, the beneficial effect of pan-HDACi relies on their capability to fine tune the miRNAs cargo of the extracellular vesicles (EVs) released by FAPs [183]. In particular, pan-HDACi treatment upregulates a subset of promyogenic miRNAs into EVs released by dystrophic FAPs, creating EVs that improve MuSC differentiation in vitro and muscle regeneration in vivo of dystrophic *mdx* mice [183].

Of note, among the pan-HDACi tested in dystrophic *mdx* mice, givinostat represents the most encouraging one to date due to its safety profile. It is a hydroxamate. The fact that givinostat has already been successfully tested in a clinical study in pediatric populations affected by systemic-onset juvenile arthritis made it possible, following promising preclinical studies, for the transfer of this drug into a clinical trial for DMD [184].

*Mdx* mice treated daily with 5 mg/kg of givinostat for 3.5 months showed histological and functional muscle improvements; the HDACi exerted numerous beneficial effects ranging from reduction of inflammation and fibrosis to promotion of skeletal muscle regeneration [163]. The beneficial outcomes of givinostat were also dependent on its effects on *mdx* muscle metabolism and mitochondrial content and quality. Indeed, mitochondrial dysfunction has been implicated as an important actor in skeletal muscle diseases, including DMD [185]. Givinostat treatment increased acetylation of the promoter of the PGC-1α gene, a key regulator of mitochondriogenesis, and therefore its expression, in *mdx* mice. As a consequence, givinostat induced a recovery of mitochondrial biogenesis and oxidative fiber type switch in *mdx* muscles, classifying it for the first time as a metabolic remodeling drug [185,186].

Givinostat was also recently used to treat muscle progenitor cells (MPCs) generated from human-induced pluripotent stem cells, increasing MPC proliferation and motility in vitro. These MPCs treated with givinostat were then transplanted into injured muscles of dystrophic nude mice, as a possible test of cell-therapy. Indeed, MPCs treated with givinostat were locally engrafted into the muscle and were able to restore dystrophin levels to reduce inflammation, necrosis, and fibrosis as well as to repopulate the MuSC niche [187]. This study suggests another strategy for the treatment of DMD based on the use of pan-HDACi.

All these studies characterized the functional, histological, and molecular beneficial effects of givinostat on a mild dystrophic phenotype, the *mdx* (C57BL10ScSn-Dmdmdx) mice. Of note, only in a recent study were the pharmacokinetic and muscle uptake properties of givinostat in *mdx* mice described, confirming a positive correlation between the doses of givinostat and the drug distribution in muscles and blood [188]. In the same study, givinostat was tested in a more severe mouse model of DMD, the D2.B10 mice. Long-term treatment with givinostat resulted in partial efficacy, improving muscle function and reducing muscle fibrosis in D2.B10 mice, although no significant effects were detected on myofiber cross sectional area or generation of myofibers [188].

In addition, different groups are focusing on studying the interactions of givinostat with other pharmacological interventions for DMD, such as steroids, to find a combinatory therapeutic strategy that successfully improves the beneficial effects on dystrophic muscles [184,188].

TSA was also tested in two different *Danio rerio* zebrafish models of DMD: the dmd morpholino (dmd-MO) knock-down model, in which an anti-sense morpholino cocktail was used to knock-down dystrophin, and the zebrafish dmd mutant line. In both models, the pan-HDACi was able to rescue muscle fiber damage [189]. The zebrafish DMD-MO model has been confirmed as a valid tool for rapid and cost-effective small molecule screening in another study. A novel chemical-combination screen of a library of epigenetic compounds identified a specific combination of the class I and II HDACi, oxamflatin, and the class III HDACi, salermide, able to ameliorate skeletal muscle phenotype in DMD mutant zebrafish, increasing the acetylation profile of histone H4 [190]. Salermide has also been described to protect muscle cells against oculopharyngeal muscular dystrophy (OPMD) in *Caenorhabditis elegans* [191].

The zebrafish model was also recently employed to study the effect of a new HDAC8 inhibitor, PCI-34051. Indeed, by using DMD patient-derived myotubes, Spreafico and colleagues observed an increase of HDAC8 activity, which was downregulated by PCI-34051 treatment [136]. Inhibition of HDAC8 in DMD-MO zebrafish treated with PCI-34051 in vivo ameliorates the dystrophic phenotype, partially repairing muscle lesions and reducing the inflammation process. At the molecular level, it was observed that the effect on the inflammation process is due to the ability of PCI-34051 to reduce IL-1b expression and, thus, immune cell recruitment. Interestingly, the beneficial effects of PCI-34051were similar to the achieved by the pan-HDACi givinostat, except for the reduction of inflammation, which was more pronounced in PCI-34051-treated zebrafish [136].

Almost ten years ago it was described how SIRT1 overexpression, in a transgenic mouse model, ameliorated the pathophysiology of DMD disease. SIRT1 overexpression decreases serum creatine kinase levels, tissue fibrosis, and myofibril damage, and increases oxidative fibers and the ability of mice to run long distances compared with control *mdx* mice [143]. SIRT1 activation improves skeletal muscle function and protects muscles of *mdx* mice by suppressing oxidative stress and also inducing the expression of antioxidative molecules such as SOD2 or catalase, acting on FoxO transcription factors [192]. For this recent evidence, SIRT1 activation is emerging as a novel therapeutic strategy for patients with MDs and drugs capable of activating the SIRT1/PGC-1α pathway may have positive effects in MD. Two small molecules were mainly used as basic direct activators of SIRT1: resveratrol and quercetin, at different doses and for different periods of treatment [193]. Resveratrol belongs to the class of flavonones and is a polyphenol compound found in foods such as grapes and red wine, and it has recently gained popularity due to its anti-inflammatory and oxidative metabolic enhancing properties [194,195]. In skeletal muscle, resveratrol may alleviate muscular dystrophic pathologies by activating the SIRT1/PGC-1α axis, therefore reducing inflammation and improving muscle function in a variety of disease models [196,197]. There are different independent studies that demonstrate how daily oral intake or intraperitoneal injections of resveratrol for several weeks in *mdx* mice contributed to the preservation of muscle function and muscle mass [198,199,200,201]. Gordon and collaborators tested, on 5-weeks-old *mdx* mice, different doses of resveratrol for a short period (10 days), and they observed with the optimal dosage (100 mg/kg) an increased expression and activity of SIRT1 and PGC-1α activation, leading to increased expression of PGC-1α and PGC-1α target genes. Unlike what chronic treatment showed in a previous study [200], this experiment also shows a reduction in the inflammatory infiltrate and increase in IL-6 [199]. Of note, the authors also observed an increase in *utrophin* gene expression. SIRT1 activation through chronic resveratrol daily administration has been shown to promote a fast to slow fiber shift in the muscle of *mdx* mice, helping the remodeling of dystrophic skeletal muscle towards a slower, more oxidative phenotype, which is known to be more resistant to the dystrophic pathology [202,203]. Long-term treatment with resveratrol also exerts beneficial effects on the hearts of *mdx* mice and of TO-2 hamster deficient in δ-sarcoglycan (LGMD2F animal model), resulting in inhibition of hypertrophy and fibrosis and improving cardiac function compared to untreated *mdx* by the downregulation of p300 protein levels and by the increase in the mitophagy process that promotes damaged mitochondrial deletion [192,204].

Similarly to resveratrol, quercetin proved to have beneficial effects against oxidative stress, neurogenic muscle atrophy [194] and DMD. Chronic quercetin dietary intake attenuates dystrophic cardio-pathology, decreasing inflammatory markers and cardiac tissue damage and increasing mitochondrial biogenesis and *utrophin* expression. Chronic treatment prevents loss of specific tension and fatigue resistance in skeletal muscle of dystrophic mice [193,205,206]. Quercetin treatment has also shown beneficial effects on dystrophic diaphragm muscles, improving respiratory function leading to an increase in the number of muscle fibers and reduced fibrotic area, but fails to increase utrophin levels, suggesting that PGC-1α/SIRT1 pathway is only partially activated in diaphragm muscle [207].

Abou-Samra et al. demonstrated that the hormone adiponectin (ApN), which has anti-inflammatory properties, is efficacious in *mdx* mice due to an effect on SIRT1 activation, which promotes upregulation of utrophin. In transgenic *mdx* mice overexpressing ApN, the investigators observed a decrease in muscle damage and enhanced muscle force compared to *mdx* mice [208].

#### 3.1.2. HDACi in Other MDs

The promising effects of using pan-HDACi have also been observed in other types of MDs (Table 2). The common pathological features of sarcolemma fragility shared by *mdx* and alpha-sarcoglycan (α-SG) null mice led to the hypothesis that pan-HDACi could be a powerful therapeutic approach also for LGMDs. Indeed, TSA treatment promotes the in vitro differentiation of MuSCs isolated from α-SG null mice. Furthermore, daily treatment of α-SG null mice with TSA induces muscle fiber size increase, while reducing fibrosis and inflammation [178]. The characterization of α-SG null mice identified numerous similarities with the *mdx* mouse, including the deregulation of NO synthesis due to the delocalization of the neuronal NO synthase to the sarcolemma, which causes changes in muscle metabolism, defects in mitochondrial biogenesis and dynamics, and modulation of HDAC activity [124,209,210]. In addition, Pambianco and colleagues demonstrated mitochondria defects in sarcoglycanopathy LGMD-2D skeletal muscle, associated with the sarcolemma instability. Indeed, LGMD-2D patients and α-SG null mice present reduced levels of PGC-1α and its target genes, reduced mitochondrial content and slow fiber-type composition and oxidative metabolism. Treatment of α-SG null mice with the pan-HDACi TSA increases the acetylation of histones within the PGC-1α promoter, thereby changing its chromatin assembly and enhancing gene expression, leading to a boost of mitochondrial biogenesis [211].

Exploiting drug screening associated with artificial intelligence-based predictive ADMET characterization of hits, givinostat was identified as a potential therapeutic drug for the sarcoglycanopathy LGMD-2D/R3, by inhibiting the autophagic pathway, likely by blocking HDAC6 activity, thereby leading to a partial α-SG protein rescue [212]. To strengthen these data, the authors also investigated the effect of another pan-HDACi, Belinostat, which induced a similar rescue of α-SG protein. Moreover, a synergistic effect was found by combining givinostat and the FDA-approved proteasome inhibitor Bortezomib, which blocks the proteasome activity and prevents the degradation of misfolded proteins: inhibition of both autophagic and proteasome pathways completely restored α-SG expression in the plasma membrane in mutant fibroblasts [212]. This evidence suggests a new therapeutic avenue for the treatment of LGMD-2D/R3, but also for other genetic diseases sharing similar protein degradation defects, as other sarcoglycanopathies.

Pan-HDACi were also exploited for the treatment of Myotonic dystrophy Type 1 (DM1), a genetic rare disease characterized by the expansion of CTG trinucleotide repeats in the 3′ untranslated region of the *DMPK* gene, that leads to the nuclear sequestration of the alternative splicing factor Muscleblind-like protein 1 (MBNL1). A flow cytometry-based screen identified the HDAC6 inhibitor ISOX and the pan-HDACi Vorinostat as modulators of MBNL1 expression. The treatment of DM1 patient-derived fibroblasts with ISOX or Vorinostat resulted in an increased MBNL1 expression, and a partial rescue of the splicing defect caused by (CUG)exp repeats [213].

Establishing the beneficial effects of pan-HDACi in rescuing skeletal muscles in multiple animal models, taken overall, these findings provide the preclinical basis for a rapid translation into clinical studies with MD patients.

Oculopharyngeal muscular dystrophy (OPMD) is caused by polyalanine expansion in nuclear protein PABPN1 [poly(A) binding protein nuclear 1] and characterized by muscle degeneration. Studies conducted on *Caenorhabditis elegans* transgenics expressing human PABPN1 with polyalanine expansion showed that deletion of *sir-2.1* (homologous of SIRT1), of the transcription factor *daf-16* (homologous of mammalian FoxO) and the *aak-2* (homologous of AMPK) rescued *C*. *elegans* adult phenotypes, whereas increasing *sir-2.1* dosage resulted detrimental [214]. Therefore, Sir2 inhibition protects against OPMD muscle pathology, whereas Sir2 activation is detrimental. This study suggested that the effect of resveratrol is context-dependent, increasing the resistance to muscle fatigue while enhancing the susceptibility to degeneration of the dystrophic muscle. The PABPN1 protein may have a role in muscle gene expression [215], and when mutated, may modify the beneficial effect of resveratrol on the control of energy metabolism in muscle leading to negative effects [214]. Based on the discovery that Sir2 inhibitors (sirtinol and splitomicin) promoted and the Sir2 activator (resveratrol) reduced muscle protection in PABPN1 nematodes, Pasco and collaborators, tested twelve SIRT1/2 inhibitors—sirtinol analogues—bearing different degrees of inhibition, for protection against mutant PABPN1 toxicity in Caenorhabditis elegans. Three compounds were highly efficient revealing a therapeutic potential for muscle cell protection in OPMD [191].

**Table 2 ijms-24-04306-t002:** Histone deacetylase inhibitors or activators in preclinical studies for the treatment of MDs.

HDAC Inhibitor	Drug	Disease	Treatment	Animal Model	Potential Mechanism	Outcomes
pan-HDACi	VPA, PhB	DMD	daily i.p. injections	C57BL10ScSn-DMDmdx/J mouse (early stages)	NA	Stable levels of CK [178]
	TSA		daily i.p. injections		Increase the expression of follistatin; FAP reprogramming with induction of MyoD and BAF60C and up-regulation of myomiR (miR-1.2, miR-133, and miR-206)	Decrease CK levels, improvement of muscle force and muscle cross sectional area, reduction of fibrosis and fat deposition, reduction of necrosis, upregulation of follistatin [168,169,178]
	SAHA		daily i.p. injections		Restoration of normal connexins (Cx): reduction of Cx40 expression and partially normalization of Cx43 distribution. Expression of Sodium Channel.	Improvement of muscle function, reduction of fibrosis and inflammation, reduction of arrhythmias [179]
	Givinostat (ITF2357)		oral administration (gavaje)		NA	Recover of muscle function, reduction of fibrosis and inflammation [163]
				C57BL10ScSn-DMDmdx/J cross with PhAM mice (C57BL6/129SV)	Activation of PGC-1α promoter	Recover of mitochondrial biogenesis, improvement of mitochondrial function, promotion of oxidative fiber type switch [186]
			oral administration (in drinking water)	D2.B10 *mdx* mouse	NA	Improvement of muscle function, reduction of fibrosis. No significant effects on CSA and on new fiber generation [188]
	Givinostat+ Steroids		Givinostat in drinking water + prednisone and deflazacort weekly i.p. injections	D2.B10 *mdx* mouse		Muscle function improvement, reduction of fibrosis; no significant effects on CSA and on new fibers generation [188]
	Givinostat		dissolved in water	dmd-MO Danio rerio zebrafish	Increase α-tubulin acetylation, decrease in il-1β expression and in the number of neutrophils/macrophages recruited at the inflammation site	Reduction of inflammation, repair of muscle lesions [136]
	Oxamflatin + Saleramide		dissolved in water	dmd-MO Danio rerio zebrafish	NA	Repair of muscle lesions [190]
	TSA	LGMD-2D	daily i.p. injections	C57BL/6 α-SG null mice	Increase of PGC-1α mRNA and protein levels and PGC-1α target genes	Increase of muscle fiber, reduction of fibrosis and inflammation, recovery of mitochondrial biogenesis and function, control of energy metabolism [178,211]
HDAC8	PCI-34051	DMD	dissolved in water	dmd-MO Danio rerio zebrafish	Increase in α-tubulin acetylation, decrease inil-1β expression and thenumber of neutrophils/macrophages recruited at the inflammation site	Repair of muscle lesions, reduction of inflammation [136]
Sir2/ SIRT1	Sirtinol	OPMD	dissolved in culture medium	PABPN1 Caenorhabditis Elegans	Modulation of the canonical Wnt signaling	Muscle protection, rescue of muscle pathology [214]
**HDAC** **activator**						
SIRT1	Resveratrol	DMD	daily dietary intake	C57BL10ScSn-DMDmdx mouse	Increase expression of PGC-1α and PGC-1α target genes	Preservation of muscle function and muscle mass, reduction of inflammation, increase of IL6, increase of utrophin, inhibition of hypertrophy and fibrosis in heart, metabolic effect on muscle fiber switch [198,199,200,201]
		LGMD2F	daily dietary intake	TO-2 hamsters, deficient in δ-sarcoglycan	Downregulation of p300 protein and increase in the mitophagy process	Inhibition of hypertrophy and fibrosis in heart [204]
	Quercetin	DMD	daily dietary intake	MDX/Utrn-/+; D2.B10-Dmdmdx/J, C57BL10ScSn-DMDmdx/J mouse	NA	Reduction of cardio-pathology, reduction of cardiac tissue damage, increase fatigue resistance in skeletal muscle, increase *utrophin* expression, increase mitochondrial biogenesis, improving respiratory function, increase the number of muscle fibers in diaphragm, reduction of fibrosis in diaphragm [193,205,206,207]
	Adiponectin		Transgenic mice with homotopic overexpression of native ApN	mdx-ApN mice	Increase of adiponectin	Marked reduction of muscular inflammation, reduction of markers of oxidative stress, decreased muscle damage, enhance muscle force, upregulation of *utrophin* [208]

NA = not addressed.

### 3.2. Clinical Trials

A high number of clinical trials involve pan-HDACi mainly in the treatment of hematologic neoplasms, but also of MDs, HIV infection, inflammatory diseases, neurodegenerative diseases, frontotemporal dementia, and Friedreich’s ataxia [184].

One of the main limitations for the use of pan-HDACi in clinics is their common adverse events, which include nausea, vomiting, anorexia, and thrombocytopenia [216,217]. Despite the positive results in preclinical studies obtained with numerous pan-HDACi for MDs, only one of them successfully arrived in a clinical trial, givinostat. The advantage of using givinostat in MD was suggested by the successful Phase I study in children affected by Systemic Onset Juvenile Arthritis (SOJIA) started in 2011 and concluded with the Phase II, in 2021, confirming that administration of givinostat (10 mg/mL oral suspension) is effective and safe [218,219] (www.Clinicaltrials.gov, clinical trial identifier: NCT00570661, accessed on 12 September 2006).

Regarding the application of givinostat in DMD treatment, in 2013 the Phase II clinical study enrolled twenty boys, aged 7 to <11 years, with a diagnosis of DMD, under stable corticosteroids regimen, and able to complete the 6 min-walk test with a minimal distance of at least 250 m. This study confirmed the ability of givinostat administration to significantly counteract DMD progression after one year of treatment, proven by an increase of myofiber cross-sectional area, muscle fiber area fraction, and a reduction of fibrosis, necrosis, and fat replacement in DMD muscles. This study also determined that givinostat was safe and tolerated. No functional improvement was observed, probably due to the small size of the cohort analyzed; however, no decline of muscle performance was noted, highlighting the fact that the compound was well tolerated [220] (www.Clinicaltrials.gov, clinical trial identifier: NCT01761292, accessed on 10 April 2013). These encouraging data permitted, in 2016, the translation of givinostat into a phase III clinical study, which is focused on long-term safety, tolerability, and efficacy of the inhibitor in a large cohort of DMD patients (www.Clinicaltrials.gov, clinical trial identifier: NCT02851797 and NCT03373968, accessed on 6 June 2017). In 2017, a phase II clinical study of givinostat for the treatment of Becker Muscular Dystrophy was initiated (www.Clinicaltrials.gov, clinical trial identifier: NCT03238235, accessed on 12 December 2017). These trials are currently ongoing.

For the many beneficial effects described above for Sirt1 activators, resveratrol could be tested on DMD patients. Only one pilot study has been conducted so far administrating resveratrol to patients with DMD, BMD, or FCMD for 24 weeks. The pilot clinical study had a limited number of patients, with three types of MDs and different clinical conditions; moreover, neither a placebo-treated group nor an untreated group could be added within the study. Nevertheless, consistent with the findings on *mdx* mice, resveratrol improved motor function and muscle power in the proximal muscles of patients under the study and creatine kinase levels decreased considerably, while longer-term administration of resveratrol may be necessary to reveal the cardiac function of resveratrol in MD patients. As adverse effects, diarrhea and abdominal pain were evaluated [221].

## 4. Conclusions

HDACs are certainly involved in the development and maintenance of muscle homeostasis in response to different insults or stimuli. Consistently, HDAC expression and activity have been found altered in MDs, suggesting a role for these enzymes in the progression of the disease. Preclinical studies showed the effectiveness of different HDACi in rescuing muscle force and morphology in MD animal models and, for some of them, the molecular mechanisms and target cells have been in part clarified. However, only the pan-HDACi givinostat has been endorsed for clinical trials in MDs, raising numerous questions about the real effectiveness of the use of these drugs on patients. Safety, dosage tolerance, and drug specificity are the main limitations associated with HDACi. Considering the specific, sometimes redundant roles, of the different HDAC isoforms, is not surprising if a pan-HDAC, which blocks the activity of numerous HDAC family members, belonging to different classes, exerts unwanted side effects. Further characterization of the kinetics of the different HDAC members in terms of expression, activity, and intracellular localization, taking into consideration the different cell types residing in striated muscles, is necessary to better define their specific involvement in MDs. Moreover, preclinical studies with isoform-specific HDACi, alone, or in combination with other FDA-approved drugs, are encouraged to propose new, more efficacious pharmacological treatments to ameliorate the burden of MDs.

## Figures and Tables

**Figure 1 ijms-24-04306-f001:**
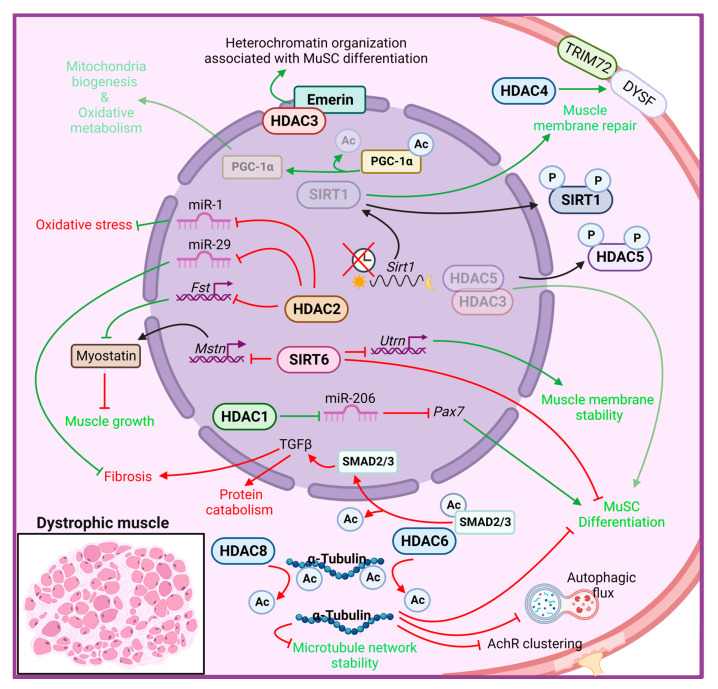
Histone deacetylase functions in muscle dystrophy condition. The cellular responses that promote MD progression are indicated in red, while in green those that counteract MD pathological features. AChR: acetylcholine receptor; Ac: acetyl group; *Fst*: follistatin; *Mstn*: myostatin; *Utrn*: utrophin; HDAC: histone deacetylase; SIRT: sirtuin; DYSF: Dysferlin; PGC-1α: peroxisome proliferator-activated receptor, gamma, coactivator 1 alpha; TGF-β: transforming growth factor beta.

**Figure 2 ijms-24-04306-f002:**
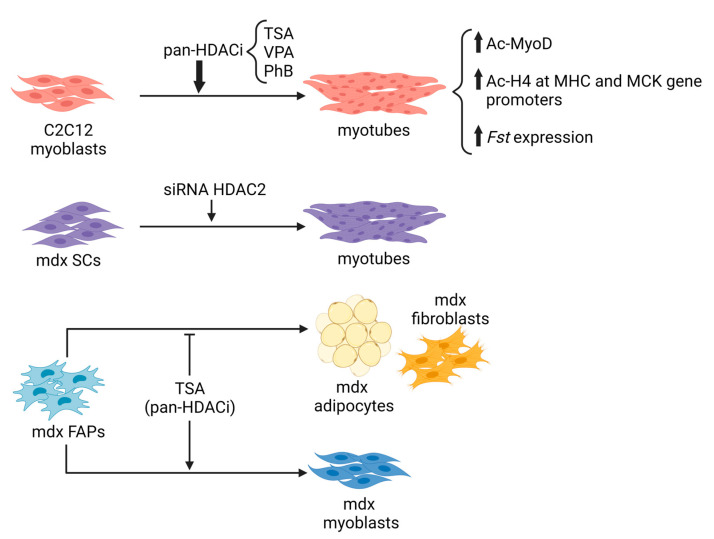
In vitro evidence of inhibiting HDACs on myoblast or FAP lineage progression. TSA: Trichostatin A; VPA: Valproic acid; PhB: Sodium Butyrate; *Fst*: follistatin; SCs: satellite cells; FAPs: Fibroadipogenic progenitors.

**Figure 3 ijms-24-04306-f003:**
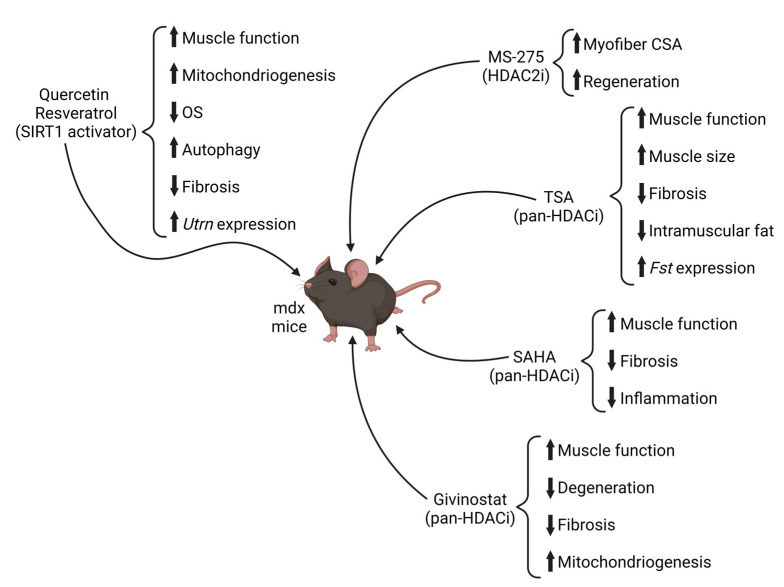
Use of HDACi in preclinical studies in DMD. *Fst*: follistatin; *Utrn*: utrophin; SIRT1: sirtuin 1; OS: oxidative stress; TSA: Trichostatin A; SAHA: suberoylanilide hydroxamic acid.

## Data Availability

Not applicable.

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
