# Peer review of "Histone Deacetylases: Molecular Mechanisms and Therapeutic Implications for Muscular Dystrophies"

_ijms, 2023, doi:10.3390/ijms24054306_

Round 1
Reviewer 1 Report
This manuscript: Histone deacetylases: molecular mechanisms and therapeutic implications for muscular dystrophies written by Alessandra Renzini at all. shows in a very clear and concise manner the underlying interaction and mechanism between enzymes that regulate the deacetylation of numerous histone and non-histone proteins, thereby affecting a wide range of cellular processes and deregulation of HDAC expression which is according to literature and this manuscript associated with several pathologies and after all suggesting potential for targeting these enzymes for therapeutic purposes.
In general, tis manuscript is review of other studies and papers but the main objective is reviewing recent insights into HDAC cellular functions in dystrophic muscles that should provide new perspectives for the development of more effective therapeutic approaches based on drugs that target these critical enzymes.
In the abstract, lines 33 and 34 authors wrote: “We will describe the signaling events that are affected by HDACs and contribute to muscular dystrophy pathogenesis by altering muscle regeneration and/or repair processes”.
That should be changed for example in to “We described” not in future tense.
Author Response
Point 1: This manuscript: Histone deacetylases: molecular mechanisms and therapeutic implications for muscular dystrophies written by Alessandra Renzini at all. shows in a very clear and concise manner the underlying interaction and mechanism between enzymes that regulate the deacetylation of numerous histone and non-histone proteins, thereby affecting a wide range of cellular processes and deregulation of HDAC expression which is according to literature and this manuscript associated with several pathologies and after all suggesting potential for targeting these enzymes for therapeutic purposes.
In general, tis manuscript is review of other studies and papers but the main objective is reviewing recent insights into HDAC cellular functions in dystrophic muscles that should provide new perspectives for the development of more effective therapeutic approaches based on drugs that target these critical enzymes.
Response 1: We thank the reviewer for appreciating our manuscript.
Point 2: In the abstract, lines 33 and 34 authors wrote: “We will describe the signaling events that are affected by HDACs and contribute to muscular dystrophy pathogenesis by altering muscle regeneration and/or repair processes”.
That should be changed for example in to “We described” not in future tense.
Response 2: We changed “we will describe” in “we describe”, since in the previous sentence the present tense has been used (Here we review..).
Reviewer 2 Report
The manuscript overviewed the family members of the Histone deacetylases (HDACs), and their functions and therapeutic implications in muscular dystrophies. It’s interesting and have important clinical application value. But there is something need to improved.
1、 The first part of “Overview of the histone deacetylase family” is too long, and there are amount of information, so it is difficult to read. The authors’ mission is not just to catalog the members of the histone deacetylase family, you should consult the literatures and summarize them. Emphasizes the valued members of histone deacetylase family.
2、 Please check whether all of references are appropriately cited, and whether have the consistent form.
3、 “4. Figures” is strange. There is only one figure, and how to associate with the other parts of the manuscript? Maybe the author should supplement more figures in paragraph of 1, 2, 3 separately. The tables are relatively clear, but if there are figures in different part, the article will be organized well.
Author Response
Comments and Suggestions for Authors
The manuscript overviewed the family members of the Histone deacetylases (HDACs), and their functions and therapeutic implications in muscular dystrophies. It’s interesting and have important clinical application value. But there is something need to improved.
Point 1: The first part of “Overview of the histone deacetylase family” is too long, and there are amount of information, so it is difficult to read. The authors’ mission is not just to catalog the members of the histone deacetylase family, you should consult the literatures and summarize them. Emphasizes the valued members of histone deacetylase family.
Response 1: We have shortened the first part (from 454 to 271 lines), focusing on the roles of HDACs in striated muscles, as target tissues of muscular dystrophies.
Point 2: Please check whether all of references are appropriately cited, and whether have the consistent form.
Response 2: We have done it.
Point 3: “4. Figures” is strange. There is only one figure, and how to associate with the other parts of the manuscript? Maybe the author should supplement more figures in paragraph of 1, 2, 3 separately. The tables are relatively clear, but if there are figures in different part, the article will be organized well.
Response 3: We have included two additional figures relative to paragraph “3. Targeting histone deacetylases in muscular dystrophies” (Figure 2) and to “3.1.1 HDACi in DMD” in preclinical studies (Figure 3).
Reviewer 3 Report
This is a well-written script. However, due to the information overload, the authors should provide additional figures, perhaps based on the categories stated in the manuscript for more clarification.
Author Response
Comments and Suggestions for Authors
Point 1: This is a well-written script. However, due to the information overload, the authors should provide additional figures, perhaps based on the categories stated in the manuscript for more clarification.
Response 1: We thank the reviewer for the positive comments. We have included two additional figures relative to paragraph “3. Targeting histone deacetylases in muscular dystrophies” (Figure 2) and to “3.1.1 HDACi in DMD” in preclinical studies (Figure 3).
Reviewer 4 Report
Renzini et al reviewed the histone deacetylase (HDAC) families and their connections with different diseases but with a focus on muscular dystrophies (MD). I like this review, it provides an overall of different classes of HDAC and its relations with diseases. I am not sure I like the idea of classifying HDAC based on its classes with diseases (Table 1), since HDAC classification has nothing to do with all different diseases. I would like to see more of which HDACs are involved in one specific disease, so it might give some clues about which HDACs could be targeted for therapeutic purposes. Overall, this review is informative and has good reference to previous work.
Some minor points to be considered:
Since there are lots HDAC and their mechanisms with MD progression or inhibition are quite different. I would suggest making a figure or table to demonstrate therapeutic features of HDAC to antagonize MD, for example which HDAC should be inhibited, which HDAC shall actually be activated. Actually, the Table 2 is good, but including an additional column describing the potential mechanism how the HDAC activation or inhibition affects the disease would be super cool.
In Figure1, please also color the lines green or red based on MD progression or MD inhibition. With so many lines and mechanisms involved, it’s good to categorize different pathways, or make them similar colors.
Author Response
Comments and Suggestions for Authors
Point 1: Renzini et al reviewed the histone deacetylase (HDAC) families and their connections with different diseases but with a focus on muscular dystrophies (MD). I like this review, it provides an overall of different classes of HDAC and its relations with diseases. I am not sure I like the idea of classifying HDAC based on its classes with diseases (Table 1), since HDAC classification has nothing to do with all different diseases. I would like to see more of which HDACs are involved in one specific disease, so it might give some clues about which HDACs could be targeted for therapeutic purposes. Overall, this review is informative and has good reference to previous work.
Response 1: We thank the reviewer for the constructive remarks. We have shortened the first part (from 454 to 271 lines), focusing on the roles of HDACs in striated muscles, as target tissues of muscular dystrophies. Therefore, table 1 has been changed, accordingly.
Some minor points to be considered:
Point 2: Since there are lots HDAC and their mechanisms with MD progression or inhibition are quite different. I would suggest making a figure or table to demonstrate therapeutic features of HDAC to antagonize MD, for example which HDAC should be inhibited, which HDAC shall actually be activated. Actually, the Table 2 is good, but including an additional column describing the potential mechanism how the HDAC activation or inhibition affects the disease would be super cool.
Response 2: We have included an additional column in table 2, describing the potential mechanisms of how HDAC activation or inhibition affects the disease. Moreover, we have included an additional Figure 3, in which we have summarized the effects of HDAC inhibitors or activators in preclinical studies on DMD.
Point 3: In Figure1, please also color the lines green or red based on MD progression or MD inhibition. With so many lines and mechanisms involved, it’s good to categorize different pathways, or make them similar colors.
Response 3: We have colored the lines in Figure 1 as requested.
Round 2
Reviewer 2 Report
The authors take care to revised the manuscript. It's good, but please sprinkled the figs into text instead of as independent apart of “4. Figures” .
Author Response
Figures have been inserted into the main text close to their first citation.